# The Potential of Lifestyle Medicine: Strategies to Optimize Health and Well-Being in Oncology Care with Dr. Amy Comander

**DOI:** 10.3390/cancers15225323

**Published:** 2023-11-08

**Authors:** Harshal Chorya, Helena S. Coloma, Viviana Cortiana, Muskan Joshi, Gayathri P. Menon, Maduri Balasubramanian, Chandler H. Park, Yan Leyfman

**Affiliations:** 1Medical College Baroda, Baroda 390018, India; 2Harvard University, Cambridge, MA 02138, USA; 3Department of Medical and Surgical Sciences (DIMEC), University of Bologna, 40126 Bologna, Italy; 4Tbilisi State Medical University, Tbilisi 0186, Georgia; 5Norton Cancer Institute, Louisville, KY 40202, USA; 6Icahn School of Medicine at Mount Sinai South Nassau, Oceanside, NY 11572, USA

**Keywords:** lifestyle medicine, cancer survivorship, cancer prevention, oncology care, health optimization, childhood cancer

## Abstract

**Simple Summary:**

This collection of discussions by an experienced speaker explores the evolving landscape of lifestyle medicine in the context of cancer care and survivorship. Emphasizing the critical role of lifestyle modifications, the discussions address challenges and opportunities in optimizing cancer survivors’ health and mitigating late effects in pediatric populations. The importance of education, patient-centered communication, and access to resources is underscored, as well as the healthcare professional’s ability to effectively support lifestyle changes. The collection also reflects on the role of nutritionists in guiding breast cancer patients considering calorie restriction. Ultimately, these insights illuminate the multifaceted nature of lifestyle medicine in cancer care and highlight the transformative power of passion and curiosity in healthcare careers.

**Abstract:**

The field of lifestyle medicine in cancer care and survivorship is undergoing significant transformation, presenting both challenges and opportunities. This collection of insights and reflections by an esteemed speaker aims to address critical facets of this evolving landscape and the intersection of healthcare, lifestyle, and cancer. With a focus on optimizing the health of cancer survivors, the speaker emphasizes the correlation between general population health and strategies for mitigating cancer risk. Evidence-based resources have a key role in their comprehensive insights into lifestyle changes’ role in cancer prevention and survivorship. Lifestyle interventions also have a promising role in mitigating the late effects in the pediatric context. Therefore, encouraging the early adoption of healthy practices in childhood cancer survivors emerges as a pivotal strategy. Furthermore, challenges in enhancing education and access to lifestyle medicine are addressed. This highlights the importance of patient-centered communication, motivational interviewing, and personalized guidance in facilitating lifestyle changes with patients. Finally, the role of nutritionists in advising breast cancer patients to consider calorie restriction to lower IGF-1 levels is explored. This collection underscores the multifaceted nature of lifestyle medicine in cancer care, highlighting challenges, opportunities, and the transformative power of passion and curiosity in shaping healthcare careers.

## 1. Introduction

The field of lifestyle medicine in cancer care and survivorship presents both opportunities and challenges. Incorporating lifestyle medicine into cancer care poses significant challenges, including limited education, time constraints, and reimbursement obstacles. Meaningful improvements in patient outcomes require systemic changes. Comprehensive solutions are, therefore, essential in advancing this field.

Through analysis of an esteemed speaker’s series of discussions, opportunities brought forth by the changing landscape of lifestyle medicine are addressed, including optimizing cancer survivors’ health, reducing late effects in pediatric populations, and improving education and access [1]. The speaker also provides insight into facilitating lifestyle changes with patients. Reports from the World Cancer Research Foundation and the American Institute of Cancer Research offer evidence-based recommendations on reducing the risk of different types of cancer through these lifestyle changes [2]. The speaker emphasizes the importance of optimizing the health of cancer survivors, especially directed toward audiences made up of oncologists or individuals aspiring to work in the field of oncology.

The speaker points out the critical intersection of healthcare, lifestyle, and cancer as a focus in care. Lifestyle medicine tools like nutrition and physical activity enhance the well-being of cancer survivors. A holistic approach to cancer management both supports survivors and potentially reduces cancer risk for the general population, aligning with improving public health outcomes.

Incorporating lifestyle medicine into cancer care requires a patient-centered approach to communication and facilitating lifestyle changes with patients, especially those dealing with pain [3]. Strategies, such as motivational interviewing, have been shown to be effective in encouraging meaningful behavioral changes. Strong communication skills are also essential in supporting patients as they pursue healthier lifestyles.

Encouraging physical activity and maintaining a healthy weight can also help childhood cancer survivors mitigate cardiovascular complications [4]. Instilling healthy habits at a young age is crucial for long-term well-being. Moreover, the strategy of calorie restriction through lifestyle medicine shows the prospect of lowering IGF-1 levels in breast cancer patients [5].

Acknowledging and addressing ethical disparities in cancer survivorship interventions is crucial for healthcare professionals to ensure equitable access and effectiveness in healthcare. By tailoring recommendations and strategies to the specific needs and challenges of patients from diverse ethnic backgrounds, healthcare providers can enhance the prospects of improved outcomes in survivorship care. This approach not only ensures inclusivity but also demonstrates the healthcare provider’s commitment to delivering patient-centered and culturally sensitive care.

The speaker’s journey toward a career in lifestyle medicine serves as a testament to how passion and curiosity can guide one’s professional choices and lead to transformative endeavors. Healthcare professionals need to recognize the impact of lifestyle-focused approaches in cancer care and survivorship. The evolving nature of medicine offers endless possibilities for pursuing better health and well-being.

## 2. Lifestyle Changes for Cancer Prevention vs. Post-Cure Recovery

The speaker’s focus on optimizing the health of cancer survivors is of paramount importance, especially considering the audience, which is largely comprised of oncologists or individuals aspiring to work in the field of oncology. They rightly emphasized the correlation between general population health and strategies for mitigating cancer risk. In this context, they recommended valuable and evidence-based resources, notably the reports from the World Cancer Research Foundation and the American Institute of Cancer Research [2]. These reports are renowned for their rigorous data analysis and in-depth meta-analyses, specifically delving into the role of lifestyle changes in reducing the risk associated with various types of cancer.

Furthermore, the speaker made a compelling point about the broader implications of the discussed tools of lifestyle medicine, such as nutrition and physical activity. These tools are not only essential for enhancing the well-being of cancer survivors but also play a pivotal role in primary cancer prevention. This underscores the versatility and far-reaching impact of lifestyle modifications in the realm of cancer care and prevention. In essence, the speaker highlighted the need for a holistic approach to cancer management, which extends from supporting survivors in their recovery journey to proactively reducing cancer risk in the general population. This approach aligns with the broader goal of improving overall public health outcomes.

## 3. Mitigating Late Effects in Pediatric Populations through Lifestyle Medicine

Upon a thorough examination of the available data, it becomes increasingly apparent that lifestyle interventions, particularly those centered around physical activity, offer a promising avenue for ameliorating the late effects experienced by pediatric populations following cancer treatment [4].

Notably, extensive research has delved into the potential benefits of encouraging physical activity among childhood cancer survivors, a strategy often implemented during the survivorship phase of care. These investigations consistently underscore the considerable advantages that regular physical activity can bestow upon this specific demographic.

Regrettably, childhood cancer survivors find themselves at a heightened risk of developing cardiovascular complications in the long term, a risk that parallels that of their adult counterparts [6]. This highlights the critical importance of instilling healthy practices from a young age. Encouraging and facilitating regular physical activity, alongside the maintenance of a healthy body weight, emerges as a pivotal strategy in this endeavor. The proactive adoption of such practices is indispensable at every juncture of a childhood cancer survivor’s journey, with the overarching goal of mitigating the enduring adverse effects associated with cancer treatment. By instilling these principles early on, healthcare professionals can significantly contribute to the overall well-being and quality of life of pediatric cancer survivors as they navigate the path to recovery and long-term health.

## 4. Challenges in Enhancing Education and Access to Lifestyle Medicine

The speaker highlighted several significant challenges related to improving education and access to lifestyle medicine, drawing from a paper by Dr. Jennifer A. Ligibel and colleagues who surveyed oncologists [7].

First, the “lack of education” issue stands out prominently. The speaker draws attention to the limited exposure that individuals receive during their medical training, from medical school to residency and fellowship, regarding essential aspects of lifestyle medicine. This gap in education encompasses a range of critical topics, including nutrition, counseling patients on obesity management, initiating and supervising exercise regimens, and promoting whole-food, plant-based diets. Addressing this education deficit is pivotal in equipping healthcare professionals with the knowledge and skills needed to integrate lifestyle medicine effectively into their practice [8].

Second, the challenge of “time constraints” within clinical settings cannot be underestimated. Physicians often find themselves immersed in a hectic schedule, where their primary focus is understandably on immediate medical interventions and treatments [9]. As a result, the limited time available for comprehensive discussions on lifestyle-related issues, despite their significant impact on cancer treatment and prevention, poses a real hurdle.

Lastly, “reimbursement” emerges as a substantial concern. While clinical guidelines emphasize the importance of exercise in cancer care, questions linger regarding who should bear the financial burden of exercise programs and related interventions. The intricacies of billing for services, such as shared medical appointments aimed at providing patient education, add complexity to the equation, potentially acting as a barrier to widespread adoption [10].

In conclusion, these challenges underscore the multifaceted nature of integrating lifestyle medicine into cancer care. Addressing the educational deficit, finding ways to navigate time constraints, and resolving reimbursement issues are vital steps toward realizing the full potential of lifestyle medicine in improving patient outcomes and overall well-being. Recognizing these challenges is the first step toward implementing effective solutions and ultimately advancing the field of cancer care.

## 5. Challenges in Facilitating Lifestyle Changes with Patients

One of the most complex aspects of working with patients to facilitate lifestyle changes is the art of effective communication. The speaker’s example illustrates this challenge vividly: a straightforward inquiry from a doctor about a patient’s exercise routine can yield a response that is far from straightforward. Patients may provide nuanced answers, such as walking when they feel like it or when the weather is favorable. In response, the doctor’s directive advice to walk for a specific duration and frequency may not be the most effective approach.

This situation underscores the need for a more patient-centered method, often encapsulated in the concept of “motivational interviewing” [11]. It involves a more nuanced and empathetic approach to eliciting behavioral change by delving into a patient’s core values and goals. Instead of simply instructing patients as to what actions they should perform, this approach encourages patients to arrive at decisions that genuinely resonate with their own aspirations and motivations.

One of the hurdles in adopting this approach is that traditional medical education often emphasizes direct and prescriptive communication. Medical professionals may not receive comprehensive training in these patient-centered techniques during their formal education [12]. Therefore, mastering the art of effectively communicating behavioral changes to patients by encompassing both coaching skills and lifestyle medicine education becomes a crucial challenge.

Furthermore, it is important to recognize that this challenge extends beyond cancer care. It pertains to a broader spectrum of healthcare, impacting individuals dealing with various health-related issues, including weight management. Enhancing healthcare providers’ communication skills, integrating lifestyle medicine education into their training, and honing their coaching abilities are fundamental steps in fostering meaningful patient engagement and supporting lasting lifestyle modifications [13]. Ultimately, this approach holds the potential to improve the overall health and well-being of patients facing diverse health challenges.

## 6. Opinions on Calorie Restriction for Lowering IGF-1 in Breast Cancer Patients

The speaker recognizes the breast cancer patient’s intention to embark on a calorie restriction regimen with the specific goal of lowering IGF-1 levels. In such situations, the involvement of a nutritionist becomes pivotal in guiding the patient through the intricate process of calorie restriction [5]. The nutritionist’s role extends beyond simple advice; they provide tailored counseling to ensure that the patient’s dietary choices encompass all essential nutrients, including proteins, carbohydrates, and fats. This holistic approach helps the patient achieve their desired outcome while maintaining overall nutritional balance and well-being.

However, it is essential to acknowledge that not all cancer centers may have readily available resources, such as nutritionists, to support patients in this manner. This inconsistency in the availability of specialized support presents a challenge that healthcare professionals may encounter. It highlights the importance of considering the healthcare setting’s resources and capabilities when planning and implementing interventions involving calorie restriction.

In summary, the speaker underscores the significance of personalized guidance and nutritional support for breast cancer patients, considering calorie restriction as a strategy to reduce IGF-1 levels. Additionally, they draw attention to the real-world challenges that healthcare providers may face in healthcare settings with varying levels of resources and support, emphasizing the need for adaptability and resourcefulness in patient care.

## 7. Personalized Comprehensive Pain Management Strategies for Enhanced Patient Recovery

Enhancing the management of pain in patients post-surgery or those grappling with pain related to underlying diseases or treatment side effects demands a comprehensive and individualized approach. Pain is a pervasive concern among patients, and addressing it effectively necessitates a multifaceted strategy [3].

The initial step in this approach involves a thorough assessment of the pain itself. Healthcare providers need to gain a comprehensive understanding of the nature and intensity of the pain experienced by the patient. This assessment forms the foundation for tailoring a suitable treatment plan. It is crucial to pinpoint and address the root causes of the pain whenever possible. Medications, such as analgesics or anti-inflammatory drugs, often play a pivotal role in pain management and can help alleviate immediate discomfort.

However, successful pain management extends beyond pharmacological interventions [14]. Many patients also experience pain in conjunction with heightened levels of stress, anxiety, and disruptions in their sleep patterns. These psychological and lifestyle factors can significantly influence the perception and experience of pain. Therefore, a holistic approach is essential, one that considers the interplay between physical discomfort and the patient’s mental and emotional well-being.

This is where lifestyle medicine tools come into play. By incorporating techniques and strategies that address stress reduction, anxiety management, and sleep improvement, healthcare providers can offer patients a more comprehensive and personalized pain management strategy. These lifestyle medicine interventions can empower patients to actively participate in their recovery and pain management process, ultimately leading to a more positive and effective outcome.

In summary, developing personalized, comprehensive pain management strategies goes beyond just treating pain symptoms with medication. It involves understanding the unique factors contributing to a patient’s pain experience, which can include physical, psychological, and lifestyle-related elements. By addressing these factors in a personalized manner, healthcare providers can optimize pain management and contribute to the patient’s overall recovery and well-being.

## 8. Ethnic Disparities in Cancer Survivorship Interventions

In the realm of cancer survivorship interventions, the critical consideration of potential ethnic disparities becomes paramount. Healthcare providers must be attuned to the diverse needs and circumstances of their patient population to ensure that interventions are not only effective but also accessible [15].

Take, for example, the promotion of physical activity as a vital component of survivorship care. Healthcare professionals should delve beyond the general recommendation and take into account the specific living conditions of individual patients. This includes assessing the safety of their neighborhoods for outdoor activities and determining whether they have access to a gym or exercise equipment at home. In circumstances where conventional exercise options may be limited, fostering creativity and collaboration with patients is essential to identify alternative approaches that suit their unique situations.

Similarly, dietary recommendations, such as increasing fruit and vegetable consumption, should be adapted to the distinct challenges faced by various ethnic groups. Access to fresh, affordable produce can be a considerable obstacle for certain patient demographics. In response, healthcare providers should explore innovative solutions, such as advocating for the use of frozen vegetables or canned beans, which may present more financially viable and accessible options. This adaptability in intervention strategies is a testament to the commitment to overcoming practical barriers that some ethnic groups may encounter in their journey toward healthier lifestyles during cancer survivorship.

In essence, the recognition and proactive mitigation of ethnic disparities in cancer survivorship interventions underscore the commitment to equitable access and effectiveness in healthcare [16]. Tailoring recommendations and strategies to accommodate the specific needs and challenges of patients from diverse ethnic backgrounds not only ensures inclusivity but also enhances the prospects of improved outcomes in survivorship care. It is a testament to the healthcare provider’s dedication to delivering patient-centered and culturally sensitive care.

## 9. Motivation and Expectations in the Pursuit of This Field of Lifestyle Medicine

Delving into the speaker’s journey and reflection on their career path, there is a resounding message of passion and the pursuit of genuine interest. The past decade has witnessed remarkable advancements in cancer treatments, which serve as a testament to the constantly evolving nature of the medical field. 

The speaker’s journey took an unforeseen turn a decade ago when she serendipitously decided to attend a lifestyle medicine conference. Back then, she carved out a single day from her hectic clinic schedule with the modest goal of exploring exercise and related subjects. Little did she foresee that this seemingly impromptu choice would mark a pivotal moment, igniting her inspiration to launch a specialized program for women with breast cancer.

This experience underscores a vital lesson in the healthcare profession: he significance of remaining open to new opportunities and being receptive to unexpected sources of inspiration. It reminds us that the journey in medicine can often be unpredictable, and some of the most profound and impactful initiatives can emerge from moments of inspiration that catch us off guard.

In essence, the speaker’s journey emphasizes the paramount importance of passion and curiosity in shaping one’s career choices. It also highlights that, at times, the most meaningful and transformative endeavors are born out of the unanticipated and the unscripted, reinforcing the notion that, in healthcare, following one’s genuine interests can lead to unexpected and extraordinary contributions to the field.

## 10. Conclusions and Future Directions

In conclusion, this exploration of lifestyle medicine’s role in cancer prevention, survivorship, patient engagement, and addressing ethnic disparities reveals both the significance and the challenges it presents. Lifestyle medicine’s potential to optimize cancer care is underscored by the speaker’s emphasis on its versatile applications in both prevention and post-cure recovery [17]. Leveraging evidence-based resources like the World Cancer Research Foundation and the American Institute of Cancer Research reports can provide invaluable guidance for healthcare professionals in the future.

Also, in the pediatric context, mitigating late effects in children through lifestyle interventions is a promising area. Future research should focus on tailored exercise and dietary programs to address the unique needs of childhood cancer survivors, ensuring their long-term well-being.

To achieve that, several challenges remain to be addressed, such as medical education, time constraints, and reimbursement. Enhanced medical curricula, innovative care models, and policy reforms can facilitate the integration of lifestyle medicine into cancer care, improving patient outcomes. Effectively facilitating lifestyle changes with patients requires refining communication through techniques like motivational interviewing. Therefore, comprehensive training that incorporates coaching skills and lifestyle medicine education can enhance patient engagement.

An additional crucial aspect will be recognizing and addressing ethnic disparities in cancer survivorship interventions. Tailoring interventions to diverse patient populations and advocating for equitable access to resources can ensure inclusive and effective survivorship care.

Therefore, to pursue lifestyle medicine’s potential, continued research, education, and policy changes are essential. This holistic approach has the power to enhance patient outcomes, reduce cancer risk, and promote a healthier society overall.

## Data Availability

Not applicable. No patient data were directly utilized in this study.

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
