# Peer review of "The Potential of Lifestyle Medicine: Strategies to Optimize Health and Well-Being in Oncology Care with Dr. Amy Comander"

_cancers, 2023, doi:10.3390/cancers15225323_

Round 1

Reviewer 1 Report

Comments and Suggestions for Authors

This manuscript is well written personal commentary.

Author Response

Thank you very much for your review and comments.

Reviewer 2 Report

Comments and Suggestions for Authors

The article is clear and well written, I have no suggestions to introduce changes and it can be published as is.

Author Response

Thank you for your review and comments.

Reviewer 3 Report

Comments and Suggestions for Authors

The authors present a very interesting issue, the one of lifestyle medicine and its benefits especially when it comes to oncological cases. The article is particularly interesting due to the fact that when it comes to oncological cases attention is rather focused strictly on medical treatments and not on the possibilities of improving the outcomes by a simple modification of pateints' lifestyle

Comments on the Quality of English Language

Minor English corrections are needed 

Author Response

(The authors gave the same response as above.)
